# Aldolase: A Desirable Biocatalytic Candidate for Biotechnological Applications

**Moloko G. Mathipa-Mdakane \* and Lucia Steenkamp**

Chemicals Cluster, Council for Scientific and Industrial Research [CSIR], P.O. Box 395, Pretoria 0001, South Africa; lsteenkamp@csir.co.za
**\*** Correspondence: mmathipamdakane@csir.co.za

**Abstract:** The utilization of chemical reactions is crucial in various industrial processes, including pharmaceutical synthesis and the production of fine chemicals. However, traditional chemical catalysts often lack selectivity, require harsh reaction conditions, and lead to the generation of hazardous waste. In response, biocatalysis has emerged as a promising approach within green chemistry, employing enzymes as catalysts. Among these enzymes, aldolases have gained attention for their efficiency and selectivity in catalyzing C-C bond formation, making them versatile biocatalysts for diverse biotechnological applications. Despite their potential, challenges exist in aldolase-based biocatalysis, such as limited availability of natural aldolases with desired catalytic properties. This review explores strategies to address these challenges, including immobilization techniques, recombinant expression, and protein engineering approaches. By providing valuable insights into the suitability of aldolases as biocatalysts, this review lays the groundwork for future research and the exploration of innovative strategies to fully harness the potential of aldolases in biotechnology. This comprehensive review aims to attract readers by providing a comprehensive overview of aldolase-based biocatalysis, addressing challenges, and proposing avenues for future research and development.

**Keywords:** biocatalysis; enzymes; aldolase; glycolysis; aldol reactions; substrate specificity

## 1. Introduction

Chemical reactions have been used for the production of certain products in different industries, including agriculture, pharmaceuticals, flavor, and fragrances. The present state of industrial production heavily depends on catalytic chemical reactions, yet this approach comes with numerous disadvantages. Research has shifted from chemical reactions to the study of green chemistry methods that are developed to promote the design and implementation of environmentally friendly chemical processes and products. Enzymes emerge as the most promising green catalyst among the range of environmentally friendly methods, facilitating a shift towards reducing hazardous waste and advancing the chemical industry's journey towards a sustainable and environmentally friendly future [1]. They operate under mild reaction conditions, typically at room temperature and neutral pH, which can significantly reduce energy consumption and minimize waste in the form of by-product production [2]. The practice of employing enzymes to catalyze reactions preventing the need for harsh chemicals or solvents in the execution of reactions is termed biocatalysis.

Biocatalysis has been shown and reported to have an advantage over chemical synthesis due to its ability to form stable intermediates, which may be devoid of toxic or pharmacological activity [3]. It is their selectivity and ability to minimize side reactions that make them particularly desirable. Additionally, this process often remains the most cost-effective path to discover new products, and hence is favored in different industries. During the past decade, various companies have extensively employed a significant number of enzymes in biocatalytic processes, including pharmaceutical and agrochemical companies [4]. These enzymes encompass a wide range of natural catalysts and possess exceptional specificity, making them highly valuable for various industrial processes [5].

Fructose 1,6-bisphosphate aldolase (FBA) is a cytosolic enzyme responsible for catalyzing the fourth stage of glycolysis [6]. The ability of aldolases to facilitate the formation of carbon–carbon (C-C) bonds and exhibit various nonenzymatic functions positions them as a compelling pathway for the synthesis of biologically significant compounds [6]. This reaction is a fundamental aspect of contemporary synthetic organic chemistry, serving as a catalyst for the conversion of FBA to glyceraldehyde-3-phosphate (G3P) and dihydroxyacetone phosphate (DHAP) within the glycolytic metabolic pathway, ultimately generating energy [7]. Figure 1 illustrates a fundamental aspect of contemporary synthetic chemistry, showcasing this catalytic reaction. The product of this reaction is a novel carbon–carbon bond, along with the potential creation of up to two new stereogenic centers [8]. Considering this aspect, this reaction has been extensively utilized in the selective synthesis of natural products and intricate bioactive small molecules with a strong emphasis on stereo control [9].

**Figure 1.** The Fischer projection illustrates the components involved in reactions catalyzed by FBA (Created with BioRender.com- https://app.biorender.com; access on 11 December 2023).

Therefore, the objective of this review is to present compelling evidence that highlights the desirability of aldolases as promising candidates for further biocatalytic investigations. It highlights existing research and studies related to aldolases as biocatalysts, providing a comprehensive overview of the field. The paper also sheds some light on their use in various industries, such as pharmaceuticals, chemical synthesis, food production, and biofuel production to help researchers and industrial practitioners understand the potential of aldolases as versatile biocatalysts in different contexts. It further analyzes the available literature to help consolidate knowledge and identify key trends, challenges, and knowledge gaps. To conduct the present study, a thorough search was performed in multiple databases (Science Direct, Google Scholar, and PubMed) to retrieve original and published scientific review articles focused on aldolases.

## 2. Taxonomy and Classification of Aldolases

Aldolase is a widely recognized enzyme classified under the lyase family, serving a vital function in carbohydrate metabolism. There are two different classes of aldolase enzymes that have been distinguished based on their mechanism and substrate specificity. Class I aldolases operate without the need for cofactors and employ a conserved lysine residue to form a Schiff base intermediate with the active carbonyl group of the donor substrate [8]. The resulting enamine attacks the acceptor substrate (the carbonyl carbon on the acceptor), resulting in the formation of the aldol product (a C-C bond) [10]. On the other hand, Class II aldolases depend on a catalytic zinc ion and exhibits metal dependency, as the

zinc ion activates the bound donor structure [11]. The resulting zinc enolate intermediate directly attacks the acceptor substrate to produce the aldol product [8]. Class II aldolases are generally reported to be more stable than Class I aldolases [12]. It should be noted that both classes of aldolases demonstrate specificity towards the donor substrate while accommodating various acceptor substrates [13]. This implies that they can tolerate a wide range of acceptor substrates while maintaining high specificity for the donor substrate [11]. Such flexibility allows them to be utilized for various applications.

Considering these two classes of aldolases, it is important to understand their distribution. Class I aldolases are typically found in higher eukaryotic organisms such as plants and animals, although studies have reported their presence in prokaryotes as well [12]. Within class I aldolases, there are three isomers: aldolase A (ALDOA), aldolase B (ALDOB), and aldolase C (ALDOC), which are encoded by distinct genes (Figure 2) [14]. These isomers use different substrates, glucose and fructose, and thus the subsequent products are different. Class II aldolases, on the other hand, are more prevalent in morphologically less complex eukaryotic organisms such as protozoans, fungi, yeasts, and algae, as well as in prokaryotes and certain archaea [15–17]. However, some organisms such as *Escherichia coli*, *Mycobacterium tuberculosis*, and *Streptococcus pneumoniae* possess both of these classes [18].

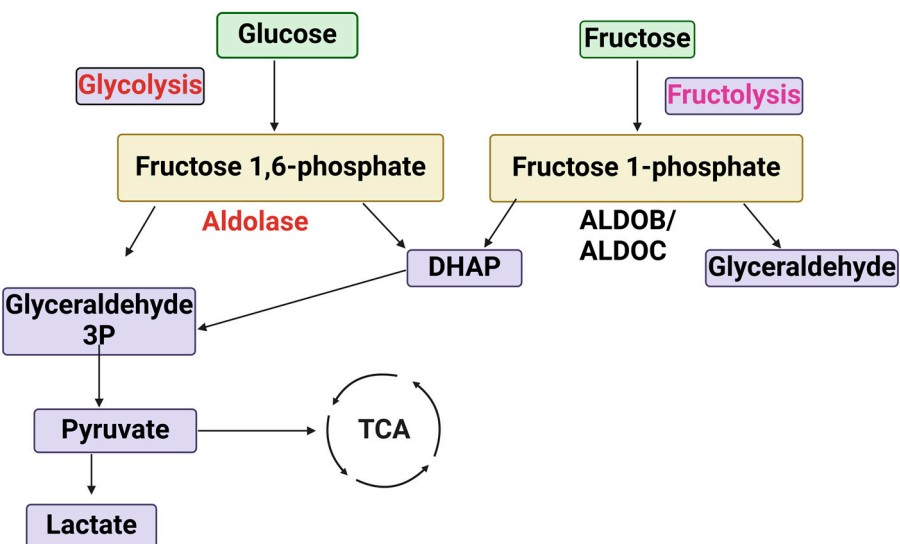

**Figure 2.** Three aldolase isomers and their respective catalytic mechanisms. This figure illustrates the catalytic mechanisms of three aldolase isomers in the breakdown of glucose and fructose during glycolysis and fructolysis. The process results in the formation of fructose 1,6-phosphate and fructose 1-phosphate. The figure highlights the specific actions of each aldolase isomer, providing insights into the crucial role these enzymes play in metabolic pathways. (Created with BioRender.com, https://app.biorender.com; access on 11 December 2023).

Classifying aldolases is crucial for understanding their biochemical functions, as well as their role in various metabolic pathways. By identifying and characterizing different types of aldolases, researchers can gain insight into the molecular mechanisms of cellular processes such as glycolysis, gluconeogenesis, and the biosynthesis of various biomolecules [19]. This knowledge can then be applied to develop novel therapeutic approaches for diseases such as diabetes, cancer, and metabolic disorders. Furthermore, the categorization of aldolases can assist in the identification and enhancement of enzymes with enhanced catalytic efficiency and specificity, offering significant implications for industrial and biotechnological applications. Comprehensive knowledge of aldolase classification provides valuable insights into their biological functions and lays the foundation for the advancement of novel enzymes and pharmaceuticals.

### 3. Elucidating the Catalytic Mechanisms of Aldolases

Aldolases play a pivotal role in glycolysis, a highly conserved metabolic pathway consisting of ten steps, wherein glucose undergoes catabolism to yield pyruvate, resulting in the generation of energy-rich molecules such as adenosine triphosphate [ATP] and reduced nicotinamide adenine dinucleotide (NADH) [6]. Additionally, they also play a role in gluconeogenesis, the metabolic pathway involved in the synthesis of glucose from pyruvate and fructose metabolism [20]. The use of aldolases as biocatalysts has gained significant attention in the flavor and fragrance industry, because of the limitations of traditional chemical production methods and due to pressure from consumers for natural products which are used on the skin or ingested. Aldolases, as biocatalysts, exhibit high specificity, leading to enhanced yields and purity of the desired product, thus minimizing the need for multiple purification steps and resulting in a more efficient process [21]. Although the initial cost of obtaining and maintaining biocatalysts may be higher than that of traditional chemical catalysts, the improved selectivity and efficiency of biocatalysis can lead to cost savings over an extended period of time [22,23].

In class I aldolases, the enzyme binds to the substrate, leading to the formation of a Schiff base intermediate between the lysine residue on the enzyme and the carbonyl group of the substrate [6]. Subsequently, a proton transfer occurs, generating an enolate intermediate. This enolate intermediate then launches an attack on the carbonyl group of another substrate molecule, resulting in the formation of the aldol product. At the end of the reaction, the Schiff base intermediate is regenerated by hydrolysis. In contrast, Class II aldolases, such as 2-deoxyribose-5-phosphate aldolase, rely on a metal ion-dependent mechanism [24]. In this mechanism, the enzyme binds to the substrate and coordinates with a metal ion, typically zinc, at the carbonyl group of the substrate [25]. This coordination serves to stabilize the enolate intermediate, which subsequently undergoes an attack on the carbonyl group of another substrate molecule to yield the aldol product [6]. The metal ion also facilitates the proton transfer step and is regenerated at the completion of the reaction.

Aldolases play crucial roles in various metabolic pathways, including glycolysis, gluconeogenesis, and the synthesis of biomolecules [19]. Understanding the mechanisms employed by aldolases is essential for understanding their biological functions and developing therapeutic approaches to target these enzymes [6].

### 4. Methods and Strategies for Aldolase Production

The production of aldolases involves various methods and strategies aimed at optimizing the yield and efficiency of these enzymes. Extensive research has been conducted on aldolases because of their crucial involvement in numerous biological processes, such as the biosynthesis of amino acids and other organic compounds. Aldolases have demonstrated significant utility as biocatalysts in a variety of chemical reactions, including the synthesis of chiral compounds, the production of biofuels, and the degradation of environmental pollutants. Their capacity for regulation based on the concentrations of reactants and products makes them highly attractive as catalysts for synthetic chemistry [25].

The production of aldolases can be achieved through traditional methods, such as the extraction of enzymes from natural sources, which include microorganisms or plant tissues. This approach has been widely recognized and utilized in various industrial and biotechnological applications. The enzyme has been reported to exhibit maximum formation during the initial stationary phase of growth, followed by a subsequent decline concurrent with the consumption of the carbon source [26]. Understanding the temporal behavior of the enzyme is critical in optimizing production processes. This insight allows us to strategically time enzyme harvests, aligning with peak activity and ensuring efficient resource utilization, thereby enhancing the overall yield and cost-effectiveness of the aldolase production process. Having this information can help in the production of aldolases; however, this natural production of aldolases will be on a small scale. Therefore, there has been an interest in exploring different other approaches to produce aldolases on a larger scale, including but not limited to cell immobilization and genetic engineering.

### 4.1. Aldolase Immobilization

To address the high production costs associated with aldolases, a cost-effective technology is required. Different aldolase immobilization methods have been explored as a potential solution in industrial applications. Figure 3 illustrates the two main methods employed, namely, physical and chemical methods. Under chemical methods, two sub-categories are depicted: adsorption and entrapment. Adsorption involves the binding of enzymes to a support material through weak interactions, while entrapment involves the physical confinement of enzymes within a matrix [27]. However, in chemical methods, the figure delineates covalent binding and cross-linking as the primary approaches. Covalent binding involves the formation of strong covalent bonds between the enzyme and the supporting material, while cross-linking entails the formation of covalent bonds between enzymes or between enzymes and the supporting material [28]. This figure serves as a visual representation of the different immobilization methods, providing a comprehensive overview of the strategies employed in enzyme immobilization.

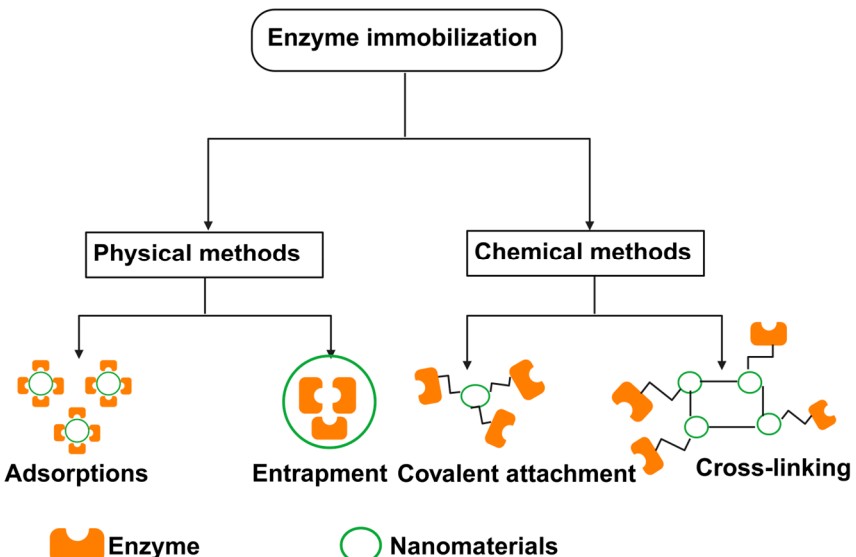

**Figure 3.** Different immobilization methods. This illustrative representation outlines two primary methods employed in enzyme immobilization, physical and chemical. Under the chemical category, the figure further elucidates the modalities of adsorption and entrapment. Simultaneously, within the realm of chemical methods, the figure distinguishes covalent bonding and cross-linking as integral subcategories. (Created with BioRender.com, https://app.biorender.com; access on 11 December 2023).

The preceding research [29] has demonstrated the feasibility of enzyme immobilization, emphasizing advantages such as reusability, easy termination of reactions, and prevention of contamination. Covalent attachment to solid supports ensures sustained catalytic binding, leading to heightened enzyme loading and activity [30]. Studies on aldolase immobilization indicate improved activity, stability, and application under non-traditional conditions, with diverse supports showing effectiveness [31,32]. Threonine aldolase in a microreactor system exhibits enhanced effects for larger volumes and cost-effective chemicals [33]. Immobilization within magnesium-aluminum nitrate layered double hydroxide yields notable reusability and activity. D-fructose-6-phosphate aldolase, immobilized on glyoxal-agarose gel, demonstrates high synthetic specific activity, improving conversion rates and reaction selectivity [34]. Various supports, including layered double hydroxides and multi-walled carbon nanotubes, enhance enzyme tolerance and reusability [35,36].

The comprehensive exploration of enzyme Immobilization, particularly aldolases, emphasizes its transformative impact on catalytic processes. The advantages, ranging from reusability and easy reaction termination to enhanced stability and activity, are evident through various studies [29,30]. The thoughtful selection of supports for aldolase immobi-

lization contributes to improved performance, enabling broader applications under various conditions [31,32]. Notable examples, such as the use of layered double hydroxides and glyoxal-agarose gel, demonstrate remarkable reusability and synthetic specific activity [29,34]. Moreover, immobilization significantly enhances aldolases' tolerance to pH and temperature variations, showcasing immense potential for industrial applications [24,29–40].

### 4.2. Genetic Modifications

With a focus on "green technology", the use of biological catalysts has gained traction over chemical alternatives. Aldolases have often been observed to exhibit a limited specificity profile, restricting the range of chiral aldol products that can be synthesized [41]. To address this limitation, various engineering approaches have been explored to overcome the scarcity of naturally occurring enzymes in many industrial reactions [22]. These approaches encompass a range of engineering methods that aim to modify the stability, substrate specificity, and stereospecificity of aldolases, thereby yielding exceptional enzymes for use in biocatalytic processes. The use of recombinant DNA technology to reduce the cost of commercial enzymes and expand their substrate selection [42]. Previous studies have shown that recombinant DNA technology can be used to clone and express the aldolase gene in a suitable host organism, such as *E. coli*, yeast, or mammalian cells [43]. Furthermore, a previous study by Windle et al. [22] has reported that for many industrial reactions there are no naturally occurring enzymes, and engineering approaches have been used to address this problem.

To develop an optimal aldolase for biocatalytic applications, the genes responsible for encoding aldolases are typically expressed in a different host system through heterologous expression. Figure 4 depicts the process of genetic modification for the production of enzymes, specifically focusing on the cloning and expression of genes for the production of enzymes. The process begins with the isolation of DNA containing the gene of interest, which is then cleaved into DNA fragments. Subsequently, the gene of interest is inserted into a plasmid, resulting in a recombinant plasmid. This recombinant plasmid is then introduced into a bacterium, where it is integrated into the bacterial genome. Subsequently, the recombinant bacteria are cultured, leading to the expression of the gene and the subsequent production of enzymes for biocatalysis. This process is of significant relevance in the genetic modification of microorganisms for the production of aldolases. Using genetic modification techniques, as illustrated in the figure, microorganisms can be engineered to produce aldolases in large quantities. This has wide-ranging implications in biotechnology and industrial processes, as aldolases are utilized in the synthesis of various compounds, including pharmaceuticals and fine chemicals. Therefore, the described process of genetic modification plays a crucial role in enabling the efficient production of aldolases, thereby facilitating their application in biocatalysis and industrial processes.

When it comes to expressing prokaryote proteins, a common practice involves employing a bacterial host, *Escherichia coli* being the most widely utilized option. This choice is mainly due to its cost-effectiveness relationship, ease of cultivation, and rapid production capabilities. In a previous study [44] the FBA aldolase gene of *Thermococcus kodakaraensis* was cloned and expressed in *Escherichia coli*, with recombinant *E. coli* showing significantly higher specific activity of FBA compared to the wild type. In a separate study, directed evolution techniques were used to generate an aldolase enzyme capable of catalyzing the formation of a 'C bond with reversed stereochemistry by introducing a reactive intermediate to the opposite diastereotopic face of the aldehyde substrate [45]. This exemplifies the potential of genetic engineering to enhance the catalytic performance of aldolases, underscoring their relevance in biocatalytic applications. Furthermore, directed evolution techniques, as demonstrated in a separate study, have been instrumental in tailoring aldolase enzymes to catalyze the formation of C-C bonds with reversed stereochemistry, showcasing the versatility and adaptability of aldolases for various biocatalytic reactions [45]. These findings collectively emphasize the significance of genetic and evolutionary approaches in enhancing the efficacy of aldolases, thus highlighting their pivotal role in biocatalysis.

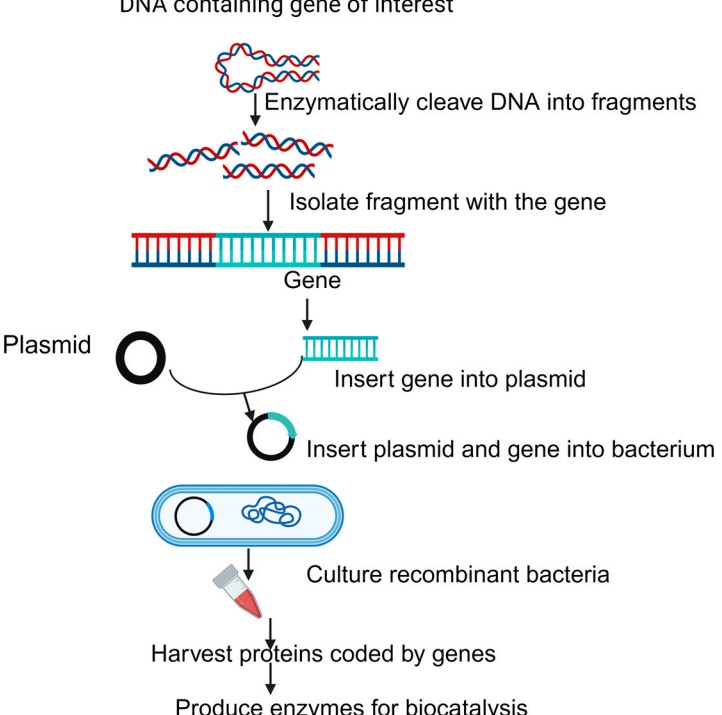

**Figure 4.** Cloning and expression of genes for enzyme production. This schematic representation illustrates the sequential steps in the genetic modification of aldolase production. With DNA cleavage, the target aldolase gene is inserted into a plasmid, forming a recombinant plasmid. After introduction into a host bacterium, cultivation results in protein synthesis, culminating in aldolase production for biocatalysis (Created with BioRender.com, https://app.biorender.com; access on 11 December 2023).

The excessive expression of aldolase in *Escherichia coli*, regulated by the T7 promoter, enhances enzyme thermostability for potential industrial use [46]. Further studies show the development of recombinant mutants through site-directed mutagenesis [47], providing insights into substrate recognition and catalytic activity [48,49]. These engineering methods, including genetic modifications, rational design, and laboratory evolution, enhance aldolase properties. After production, chromatography and purification techniques extract and purify the host cell. Recombinant technology optimizes aldolase production, offering advantages in yield and purity for industrial and biomedical applications. Engineering techniques also modify natural aldolases, enhancing their catalytic properties for industrial use [43,50]. Directed evolution, rational design, and de novo design refine and optimize aldolases, improving performance and functionality [51,52]. These studies highlight the strategies and applications of aldolases in biocatalysis. Recombinant technology's application in aldolase production signifies a significant field advancement, showcasing the efficacy of genetic engineering. Understanding these methods is crucial, emphasizing the potential benefits of using recombinant technology for aldolase production, meeting specific industrial needs, reducing costs, and optimizing enzyme production in microbial hosts. The versatility of recombinant enzymes highlights the potential to enhance aldolase production and expand their industrial applications, contributing to the advancement of enzymatic processes and biotechnology.

## 5. Industrial Applications of Aldolases as Biocatalysts

Aldolases have been used to create biosynthetic pathways for the production of valuable compounds. The enzyme's capacity to facilitate the transformation of FBA into DHAP and G3P has made it an attractive focal point for diverse sectors, such as food and pharmaceuticals. In the food industry, aldolase has been used to improve the quality of processed foods, while in the pharmaceutical industry, it has been used as a drug target

for the treatment of various diseases. Furthermore, the use of aldolase in biofuels and bioremediation applications has gained considerable interest in recent years. Figure 5 illustrates the versatile application of aldolases in different industrial sectors. Categorically, the figure specifies five primary industries, namely food, pharmaceutical, chemical, biofuel, and environmental sectors, wherein aldolases serve as potent biocatalysts. This emphasizes the adaptability of aldolases, showcasing their utility in facilitating various enzymatic reactions and processes across a spectrum of industrial domains.

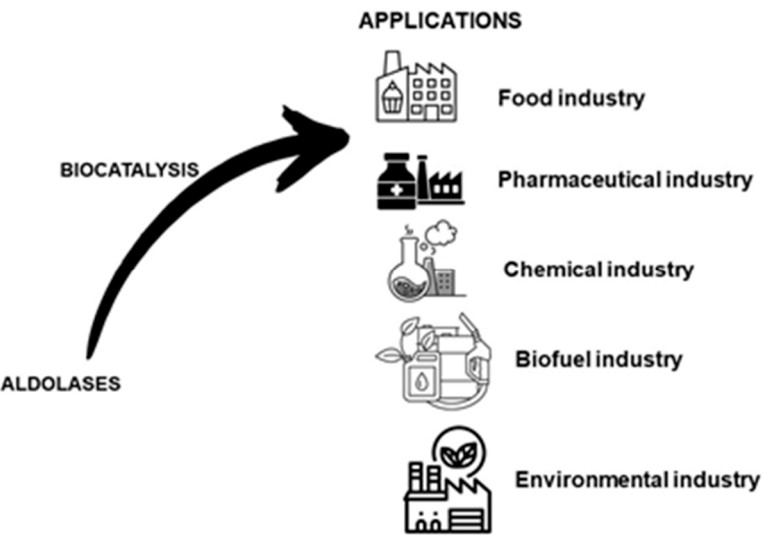

**Figure 5.** Diverse industrial applications of aldolases as biocatalysts. This illustrates the diverse industrial applications of aldolases, showcasing their potential utilization across various sectors, including the food, pharmaceutical, chemical, biofuel, and environmental industries.

### 5.1. Aldolases in the Food Industry

The utilization of aldolases in the food industry is extensive, primarily due to their capacity to catalyze the conversion of carbohydrates into valuable products. In particular, aldolases play a crucial role in the production of high-fructose corn syrup (HFCS), a widely used sweetener in the food and beverage industry known for its cost-effectiveness and improved solubility compared to conventional sugar [53]. The enzymatic conversion of glucose to fructose, a key step in the production of HFCS, is facilitated by aldolases, contributing to the reduced environmental impact of the food industry by decreasing the dependence on traditional sugar sources [54]. This process involves the use of glucose isomerase to convert glucose into fructose, followed by the utilization of aldolase to convert the resulting ketose into FBA, which can then be further converted to fructose through a series of enzymatic reactions [53,54]. The pivotal role of aldolases in the production of HFCS deepens their importance as biocatalysts in the food industry, offering sustainable and efficient pathways for the generation of valuable carbohydrate-derived products.

Aldolases have demonstrated significant utility in the synthesis of food additives and flavor compounds, contributing to the production of chiral intermediates such as β-hydroxy-α-amino acids, which are crucial precursors for flavor compound [55]. Furthermore, aldolases have been instrumental in the production of lactulose, a widely used prebiotic in the food industry, synthesized through the reaction between lactose and fructose [56]. The utilization of aldolases in the production of food additives and flavor compounds offers several advantages over conventional chemical methods, including enhanced selectivity and efficiency. Furthermore, aldolases operate under mild reaction conditions, preserving the delicate flavors of food compounds, thus supporting the development of sustainable food additives and flavor compounds.

Moreover, aldolases are involved in the production of fructo-oligosaccharides, serving as prebiotic dietary fibers with the potential to be utilized as a sugar substitute. In particular,

the enzyme threonine aldolase plays a crucial role in the breakdown of threonine in lactic acid bacteria to glycine and acetaldehyde, a significant flavor component of yogurt [57,58]. In a previous study, the catabolism of threonine was examined in *Lactococcus lactis*, and researchers identified threonine aldolase as the key enzyme that initiates the conversion of threonine to glycine, representing the primary route of threonine degradation [57].

### 5.2. Aldolases in the Pharmaceutical Industry

The adaptability of aldolases as biocatalysts in the pharmaceutical industry is due to their ability to catalyze various aldol reactions, particularly in the synthesis of crucial intermediates such as chiral β-hydroxy ketones, which play a significant role in the development of drugs such as statins and protease inhibitors [59]. Aldolases offer numerous advantages over conventional chemical catalysts in pharmaceutical synthesis, including a higher selectivity and efficiency, as well as an ability to operate under mild reaction conditions, thereby minimizing the formation of undesired byproducts. The utilization of aldolases in pharmaceutical synthesis not only enhances the sustainability of the industry by reducing the need for hazardous chemicals and solvents but also contributes to the development of environmentally friendly processes. Recent advances in protein engineering techniques have facilitated the creation of innovative aldolase biocatalysts with enhanced properties, including improved stability and activity, leading to higher yields and improved quality in the synthesis of pharmaceutical compounds.

Aldolases play a pivotal role in the pharmaceutical industry, particularly in the biosynthesis of natural products and in the synthesis of various drugs, including antiviral drugs, antibiotics, and anticancer drugs. They have been utilized in the production of macrolide antibiotics, polyketides, and other natural products, offering several advantages over traditional chemical approaches and leading to the creation of new drugs with enhanced therapeutic properties [60]. In particular, aldolases have been identified as important enzymes in the stereospecific synthesis of the central frameworks of islatravir and zanamivir, both of which are significant drugs for the treatment of HIV infection and the prevention and treatment of influenza, respectively [61]. Additionally, aldolases have been instrumental in the production of vidarabine, a drug used in the treatment of infections caused by herpes simplex and Varicella zoster viruses [62]. Furthermore, *N*-Acetylneuraminic acid aldolases have been reported to be promising scaffolds in the development of anti-influenza drugs, with successful utilization in the production of sialic acid analogs, serving as potential sources of antiviral agents [63,64].

Aldolases offer an advantage in organic synthesis by allowing reactions to occur in aqueous solutions without the requirement of protection of the hydroxyl group, unlike chemical reactions [65]. This enzymatic approach has been successfully applied in the preparation of biologically significant molecules, serving as valuable building blocks in the pharmaceutical industry. In particular, aldolases have been used in the synthesis of lipid Mycestericin D, known for its immunosuppressive activity [66,67].

Aldolases have been recognized as essential components in the synthesis of β-hydroxy-α-amino acids, widely used as active pharmaceutical ingredients (API) and potential drug candidates [67,68]. These enzymes have been identified, cloned, and purified for their role in the synthesis of β-hydroxy- α-amino acids, highlighting their importance as biocatalysts in this reaction [68]. Furthermore, aldolases have been reported to play a crucial role in the synthesis of Zanamivir, a neuraminidase inhibitor effective against influenza A and B viruses, and the side chain of Atorvastatin, a cholesterol-lowering medication [67,69–71].

### 5.3. Aldolases in the Chemical Industry

Aldolases have also emerged as very useful catalysts for chemical synthesis within the chemical industry, and their utility extends beyond facilitating aldol condensation reactions to demonstrate efficacy in chiral compound synthesis. Enantioselective synthesis, integral to pharmaceutical and agrochemical production, relies on the stereochemistry of compounds, a facet highlighted previously [72]. In this context, aldolases serve as catalysts

in asymmetric aldol reactions, ensuring the synthesis of chiral compounds with exceptional stereoselectivity. The advantageous attributes of aldolases in enantioselective synthesis, such as the ability to work in aqueous environments, mild reaction conditions, and high selectivity, as described [73] emphasize their importance in chemical processes. Furthermore, the continual evolution of aldolase biocatalysts has facilitated the production of chiral compounds with remarkable optical purity [22]. This progress not only enhances the efficiency of chemical synthesis, but also aligns with the industry's demand for high-quality, stereopure compounds. The utilization of aldolases in the chemical industry is a testament to their multifaceted role in enhancing the synthesis of chiral compounds. The demonstrated advantages, encompassing operational adaptability and superior stereoselectivity and the complexities of chemical synthesis, position aldolases as indispensable catalysts and contribute significantly to the refinement of chemical processes within this industrial domain.

The significance of aldol reactions in the production of carbohydrates, chiral building blocks, and natural polyketide products has been well documented in the literature [9,74,75]. For instance, DERA (2-deoxyribose-5-phosphate aldolase) has been used to generate chiral lactol intermediates essential for the synthesis of optically pure superstatins, such as rosuvastatin and pitavastatin, which are widely used in the pharmaceutical industry [76]. Furthermore, aldehydes, due to their distinctive reactivity, offer significant potential as platform chemicals to produce a wide range of bioproducts [74]. Research has shown the effectiveness of aldolases in catalyzing the condensation of aldehydes and/or ketones into β-hydroxyaldehydes, which can also undergo conversion to value added chemicals, including precursors for cholesterol-lowering drugs [7]. Additionally, aldolases have been shown to play a crucial role in the targeted synthesis of unusual deoxysugars and chiral intermediates through direct biocatalytic processes, highlighting their potential in the production of chiral compounds within the chemical industry [77]. Furthermore, the use of aldolases in the production of compounds such as cinnamaldehyde, renowned for its taste and aroma, further points out its significance in the chemical industry [78]. This compound is widely employed as a safe food and flavor additive, finding extensive use across a range of commercial food applications. The diverse studies presented here collectively emphasize the significant potential of aldolases as biocatalysts to produce chiral compounds within the chemical industry.

### 5.4. Aldolases in the Biofuels Industry

The utilization of aldolases in the biofuels industry has attracted considerable attention due to their ability to facilitate the conversion of FBA into dihydroxyacetone phosphate and G3P, which are crucial intermediates in the production of biofuels such as ethanol and butanol from lignocellulosic biomass [79]. The study by [79] demonstrated the potential of aldolases in the biofuels industry, highlighting their ability to catalyze key reactions in the conversion of biomass-derived substrates into biofuels. The enzymatic conversion mentioned previously in this section represents a promising pathway for sustainable biofuel production. Furthermore, the incorporation of aldolases into biofuel production processes can contribute to minimizing the environmental footprint of the industry by mitigating greenhouse gas emissions and minimizing waste generation. Enzymatic conversion of biomass-derived substrates into biofuels using aldolases offers numerous advantages, including improved selectivity, reduced energy requirements, and environmental benefits. The relevance and potential of aldolases for the advancement of the biofuel industry toward more sustainable and environmentally friendly practices is highlighted.

Aldolases have emerged as valuable tools in the advancement of innovative biorefinery processes aimed at biofuel production. These processes involve the conversion of lignocellulosic biomass into various high-value outputs, including biofuels, chemicals, and materials [80]. Aldolases have been instrumental in the synthesis of platform chemicals such as hydroxymethylfurfural, levulinic acid, and 3-hydroxypropanoic acid, which serve as crucial intermediates in the production of biofuels and other chemicals [81,82]. The

integration of aldolases into biorefinery processes has the potential to contribute to the advancement of a sustainable bioeconomy that relies on the efficient utilization of renewable resources for the production of various goods [80]. This enzymatic approach presents a promising pathway for sustainable biofuel production, contributing to minimizing the environmental footprint of the industry by mitigating greenhouse gas emissions and waste generation [83]. In conclusion, the integration of aldolases into biorefinery processes for biofuel production represents a significant advancement towards a sustainable bioeconomy. As the demand for renewable energy sources continues to grow, the role of aldolases in biofuel production is poised to become increasingly significant, contributing to the development of a more sustainable and environmentally friendly energy landscape.

### 5.5. Aldolases in the Environmental Industry

Aldolases have found applications in the environmental sector, particularly in the production of polyhydroxyalkanoates (PHA) from sustainable sources such as lignocellulosic biomass and waste materials. These compounds serve as crucial building blocks in the manufacturing of biodegradable plastics, which can replace traditional petroleum-based plastics, thereby reducing the environmental impact of plastic waste [84,85]. The use of aldolases in the production of PHA aligns with the global focus on developing environmentally friendly products, including biofuels, biochemicals, bioplastics, and biocomposites, from natural biomass [86]. The biodegradability of PHAs, facilitated by aldolases, offers a promising solution to the challenges posed by conventional plastics, contributing to the development of sustainable materials for various applications, including food packaging and agricultural waste management [87].

Aldolases have been used in the bioremediation of contaminated environments, particularly in the degradation of aromatic compounds such as benzoic acid, phenol, and catechol, which are common pollutants in the environment [88]. The use of aldolases in bioremediation offers several advantages over traditional chemical methods, and advancements in protein engineering techniques have facilitated the creation of innovative aldolase biocatalysts with enhanced stability and activity, enabling the efficient degradation of toxic substances present in polluted environments [22]. This application of aldolases in bioremediation is consistent with the global focus on developing sustainable solutions to environmental pollution. Moreover, the potential for aldolases to contribute to a microplastic-free environment and mitigate the environmental impact of plastics pollution has been highlighted in the literature.

The utilization of aldolases in the environmental industry to produce biodegradable plastics from renewable sources represents a significant step towards addressing the environmental challenges associated with conventional plastics. As the demand for sustainable materials continues to grow, the application of aldolases in the environmental sector is poised to play a crucial role in promoting environmentally friendly and sustainable solutions for various industries.

The versatile application of aldolases in various industrial processes underscores their significance as powerful biocatalysts. From pharmaceuticals to fine chemicals, aldolases play a pivotal role in synthesizing valuable compounds with high efficiency and specificity. Table 1 provides a comprehensive summary of these industrial applications, highlighting the diverse substrates, products, and advantages associated with aldolase-mediated reactions.

**Table 1.** Summary of industrial applications of aldolases as biocatalysts.

| Industry | Uses | References |
| --- | --- | --- |
| Food | Production of high-fructose corn syrup (HFCS), a widely used sweetener in the food and beverage industry | [53] |
| | Used in the synthesis of food additives and flavor compounds, which are crucial precursors for flavor compound | [55] |
| | Production of lactulose, a widely used prebiotic in the food industry | [56] |
| | Play a crucial role in the breakdown of threonine in lactic acid bacteria to glycine and acetaldehyde, a significant flavor component of yogurt | [57,58] |
| Pharmaceutical | Play a significant role in the development of drugs such as statins and protease inhibitors | [59] |
| | Utilized in the production of macrolide antibiotics, polyketides, and other natural products, leading to the creation of new drugs with enhanced therapeutic properties:<br>• islatravir and zanamivir, significant drugs for the treatment of HIV infection and the prevention and treatment of influenza<br>• vidarabine, a drug used in the treatment of infections caused by herpes simplex and Varicella zoster viruses | [60–62] |
| | Used in the synthesis of lipid Mycestericin D, known for its immunosuppressive activity | [63,64] |
| | Essential components in the synthesis of β-hydroxy- α-amino acids, widely used as active pharmaceutical ingredients (API) and potential drug candidates | [66,67] |
| Chemical | Catalysts in asymmetric aldol reactions, ensuring the synthesis of chiral compounds with exceptional stereoselectivity | [72] |
| | Facilitate the production of chiral compounds with remarkable optical purity | [21] |
| | Generate chiral lactol intermediates essential for the synthesis of optically pure superstatins which are widely used in the pharmaceutical industry | [76] |
| | Play a crucial role in the targeted synthesis of unusual deoxysugars and chiral intermediates through direct biocatalytic processes | [77] |
| Biofuels | Catalyze key reactions in the conversion of biomass-derived substrates into biofuels | [79] |
| | Conversion of lignocellulosic biomass into various high-value outputs, including biofuels, chemicals, and materials | [80] |
| | Instrumental in the synthesis of platform which serve as crucial intermediates in the production of biofuels | [81,82] |
| Environmental | Production of polyhydroxyalkanoates (PHA) for building blocks in the manufacturing of biodegradable plastics | [84–86] |
| | Used in the bioremediation of contaminated environments | [88] |

## 6. Challenges Associated with Aldolase Based Biocatalysis

Aldolases have emerged as highly promising biocatalysts for aldol reactions, demonstrating remarkable capability in catalyzing these reactions and generating valuable organic compounds. Extensive research has highlighted the potential of aldolases in biocatalysis. However, despite the numerous benefits associated with the use of aldolases as biocatalysts, there are notable challenges specific to these enzymes. Notably, not all substrates utilized in aldolase-based biocatalysis readily dissolve in aqueous solvents, and the cost and stability of these enzymes, along with their substrates, present significant challenges [89]. Furthermore, while the advantages of the specificity of aldolases have been extensively documented, this specificity can also limit the range of available substrates and stereochemical outcomes, hindering the widespread adoption of aldolases as versatile catalysts for asymmetric synthesis [22,89]. The challenges associated with aldolase-based biocatalysis emphasize the need for further research and development to address these limitations. Efforts to enhance the solubility of substrates in aqueous solvents and improve the stability and cost-effectiveness of aldolases are crucial in expanding their applicability in biocataly-

sis. Additionally, exploring strategies to broaden the substrate scope and stereochemical outcomes of aldolases can contribute to overcoming specificity-related challenges, thereby enhancing their utility as versatile catalysts for asymmetric synthesis.

Aldolases play a crucial role in biocatalysis, offering the potential for the synthesis of a wide range of compounds. However, a significant challenge in aldolase-based biocatalysis is the limited substrate scope of natural aldolases. This limitation arises from the specificity of aldolases towards a limited selection of substrates, as well as the structural constraints of the enzyme [49]. The stringent requirements of the active sites of aldolases have been identified as a key factor contributing to the limited substrate acceptance by these enzymes [89]. Research has highlighted the specificity of aldolases toward their respective donor substrates, which prevents them from accepting other donors even if their structures are similar to the natural donor [89]. For example, DERA has been identified as catalyzing the addition of two acetaldehyde molecules to the acceptor aldehyde, resulting in G3P, in an aldol reaction involving two aldehydes [90]. On the other hand, FBA utilizes DHAP as its donor substrate but can catalyze reactions with a relatively broad range of acceptor aldehydes [91]. The limited substrate scope of natural aldolases has prompted extensive research to enhance their utility in biocatalysis. While natural aldolases exhibit specificity towards a limited selection of substrates, ongoing research in protein engineering and immobilization techniques holds promise for expanding the substrate scope of aldolases in biocatalysis. These advancements are crucial for unlocking the full potential of aldolases in the synthesis of various compounds, thereby contributing to the advancement of biocatalysis in various industrial and pharmaceutical applications.

The stability and activity of aldolases under process conditions present significant challenges. Industrial processes often involve harsh chemical conditions such as elevated temperatures, extreme pH values, and the presence of organic solvents, which can adversely affect the stability and catalytic activity of aldolases, leading to reduced efficiency and shorter lifetimes [92]. Enzymes used in biocatalysis should be stable and resistant to organic solvents and subsequent process-related processes [93]. Aldolases are known to be costly and rather unstable, leading to the creation of variants that can accept them as alternative donors [94]. The catalytic efficiency of aldolases decreases with increasing temperatures, while their stability increases with the optimal temperature for the growth of the source organism [95]. Furthermore, aldolases exhibit low activity and stability under high substrate load conditions, limiting their potential for industrial applications [96]. Harsh industrial conditions lead to reduced efficiency and shorter lifetimes. Addressing the challenges associated with the stability and activity of aldolase enzymes is crucial for their successful application in industrial processes. Further research and development efforts are needed to enhance the stability and activity of aldolases under challenging process conditions, thereby unlocking their full potential as biocatalysts.

Scaling up aldolase-based biocatalytic processes from the laboratory to the industrial scale presents significant challenges. The importance of considering factors such as mass transfer limitations, reaction kinetics, and separation of products from reaction mixtures when working with aldolases has been well highlighted [97]. Furthermore, the integration of processes with other steps in the synthesis pathway and the advancement of continuous-flow processes add complexity to the scaling up of aldolase-based biocatalysis. A study by Timson [98] discussed challenges and strategies for the scale-up of biocatalytic processes, highlighting both challenges and suggested mitigations [98]. Studies by Schmidt [99] emphasized the crucial need to upscale the aldolase process due to the limitations of growing microorganisms in the laboratory for industrial scales. Challenges associated with process optimization, reactor design, and efficient downstream processing techniques that can negatively impact the scaling process when working with aldolases were highlighted in another study [22]. To address these challenges, the exploitation of mathematical models for the optimization of both the process and reactor design in an effort to successfully scale up enzyme production was reported [100]. A previous study [98] further highlighted the challenges associated with better biocatalysis, but also suggested that addressing these

challenges could lead to an increase in the number of successful biocatalysis projects, unlocking the full potential of aldolases as valuable tools in biocatalysis. The challenges associated with scaling up aldolase-based biocatalytic processes are multifaceted, including limitations in mass transfer, reaction kinetics, product separation, integration with other synthesis pathway steps, continuous-flow processes, and the limitations of laboratory-scale microorganism growth. However, by addressing these challenges through strategies such as process optimization, reactor design, efficient downstream processing techniques, and the exploitation of mathematical models, the full potential of aldolases can be unlocked as valuable tools in biocatalysis.

## 7. Strategies to Overcome Challenges in Aldolase Based Biocatalysis

The substrate scope limitation of natural aldolases poses a significant challenge in biocatalysis. Protein engineering emerges as a promising strategy to overcome this limitation by customizing enzyme activity, substrate selectivity, and stability for industrial processes [22]. This involves modifying enzyme properties through genetic, chemical, or computational methods. Combining biocatalysts with engineered or promiscuous enzymes allows for the expansion of synthetic cascades and the creation of novel target structures [101]. Diversification of substrate scope is achieved through protein engineering, showcasing successful modification of D-fructose-6-phosphate aldolase (FSA) from hydroxyacetone to dihydroxyacetone, enhancing substrate range and stereoselectivity [102]. Structure-based substrate design and site-specific mutation techniques are explored to expand the substrate scope, as demonstrated in the case of DERA [103]. Previous studies show a significant expansion of the substrate scope of *E. coli*'s FSA, evolving from proficiency with hydroxyacetone to accepting a wide range of alternative nucleophiles [103–105].

Various strategies, including immobilization techniques, co-factor regeneration systems, and the use of innovative support materials, address challenges associated with aldolase stability in biocatalysis [34]. Cofactor regeneration systems play a crucial role in sustaining enzymatic activity during aldolase-based processes [35,40,106]. The use of multiwalled carbon nanotubes as a support material for immobilized aldolases enhances catalyst versatility [40]. Recombinant DNA technology facilitates the production of aldolases in significant amounts, addressing availability and cost concerns [107–109]. High-throughput screening methods aid in the discovery of novel aldolases with enhanced properties [107–110].

A comprehensive strategy encompassing protein engineering, enzyme stabilization, and efficient production processes is essential for overcoming challenges in aldolase-based biocatalysis. Protein engineering techniques, including rational design and directed evolution, significantly contribute to expanding substrate scope and catalytic efficiency [73]. Stabilization strategies, such as immobilization and cofactor regeneration, enhance aldolase robustness for industrial-scale processes [111]. Efficient production processes and screening methods ensure the availability of aldolases for various biocatalytic applications [112]. Scale-up strategies, including process optimization and reactor design, are critical for transitioning aldolase-based biocatalytic processes to large-scale industrial applications [113].

Addressing the substrate scope limitation of natural aldolases in biocatalysis demands a multifaceted approach, as highlighted in this subsection. Protein engineering emerges as a pivotal strategy, allowing for the customization of enzyme activity, substrate selectivity, and stability to meet the demands of industrial processes. The successful modification of FSA exemplifies the potential of protein engineering, demonstrating enhanced substrate range and stereoselectivity. Furthermore, strategies to enhance stability and activity, including various immobilization techniques and co-factor regeneration systems, prove crucial in overcoming challenges associated with aldolase stability during biocatalysis. Efficient enzyme production through recombinant DNA technology and high-throughput screening methods addresses concerns of availability and cost-effectiveness. Embracing these diverse approaches not only expands the substrate scope but also contributes to the overall success of aldolase-based biocatalysis in industrial applications [73,111–113].

## 8. Conclusions

Aldolases, as biocatalysts, have garnered significant attention because of their potential applications in the synthesis of valuable compounds. Their significance lies in their ability to catalyze a diverse array of reactions with exceptional enantioselectivity and regioselectivity, making them valuable biocatalysts in various industrial and synthetic organic chemistry applications. However, the limited availability of suitable aldolases has hindered their full potential in biocatalytic processes. To address this limitation and propel the field forward, several recommendations for future research can be considered.

Efforts should be directed towards expanding the substrate scope of aldolases to efficiently convert a broader range of molecules, including challenging and structurally diverse substrates. This can be achieved through screening for natural sources, employing enzyme engineering techniques to enhance substrate binding, or exploring the potential of novel aldolase variants. By broadening the selection of substrates, aldolases can unlock new possibilities for their application in various biotechnological processes. The application of protein engineering techniques holds immense potential for tailoring aldolases for enhanced properties, such as catalytic efficiency, stability, and altered substrate specificity. Future research should prioritize the design and engineering of aldolases to enhance their industrial applicability. This involves developing strategies to engineer existing aldolases or create new variants with improved properties, which include stability, activity, and substrate promiscuity. Rational design based on structural information, directed evolution approaches, and computational methods can be employed to achieve these goals, contributing to the development of customized aldolases with superior biocatalytic properties.

The translation of aldolases from laboratory-scale processes to industrial applications presents challenges that necessitate future research attention. Prioritizing process optimization is crucial, including the advancement of efficient reaction conditions, innovative immobilization techniques, and the implementation of continuous-flow systems. Scale-up strategies for the large-scale production of aldolases should be investigated, and the integration of process engineering and biocatalysis expertise is vital to unlocking the full potential of aldolases in extensive biotechnological applications. To enable the practical implementation of aldolase-based biocatalytic processes, research should focus on process integration and scale-up considerations. This involves the development of effective reactor systems, fine-tuning of reaction conditions, and adoption of continuous-flow processes. The application of process engineering and design principles ensures efficient mass transfer, improved productivity, and seamless integration with downstream processing steps.

In summary, future research should strategically address the expansion of the aldolase substrate scope, the application of enzyme engineering techniques, and the optimization of processes for large-scale biotechnological applications. When these research areas are addressed, the full potential of aldolases as versatile biocatalysts can be unlocked, paving the way for their widespread adoption in various industrial sectors, including pharmaceuticals, fine chemicals, and biofuels.

**Author Contributions:** M.G.M.-M.: Conceptualization; writing- original draft preparation; M.G.M.-M. and L.S.: Writing—review and editing; M.G.M.-M. and L.S.: Contributed to the article and approved the submitted version. All authors have read and agreed to the published version of the manuscript.

**Funding:** This review was supported by the National Research Foundation [NRF], South Africa, grant ID 144837 and the Council for Scientific and Industrial Research [CSIR] parliamentary grant [PG].

**Data Availability Statement:** No new data were created or analyzed in this study. Data sharing is not applicable to this article.

**Conflicts of Interest:** The authors declare no conflicts of interest.

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
