# Peer review of "Aldolase: A Desirable Biocatalytic Candidate for Biotechnological Applications"

_catalysts, doi:10.3390/catal14020114_

Round 1
Reviewer 1 Report
Comments and Suggestions for Authors
Please see the attached file.

Reviewer 2 Report
Comments and Suggestions for Authors
The paper focuses aldolase enzymes and their importance in organic synthesis - both currently and also in the future. A special emphasis is given to the potential opportunities for scale up and production of both the enzyme and the application for making various valuable products such as pharmaceutical actives. According to the authors, this points out the huge benefits that are within reach when relying on enzymatic and specifically aldolase catalysed transformations.
The topic of the paper is huge and diverse and, unfortunately, the authors have not succeeded to entirely grasp all aspects in a succinct and understandable way. In the attached file a number of comments have been inserted to point out parts that have to be addressed, not least the author's weakness to repeat statements and messages several times. It is a strong recommendation to proof-read the manuscript and eliminate unnecessary duplications and repeats. - this only sisks that the reader will loose interest or even stop reading.
Finally, the list of references has to be tiedied up to follow a common house style.

The Enlish language is entirely acceptable, but should be shortened in specified parts.
Round 2
Reviewer 1 Report
Comments and Suggestions for Authors
Accept in the current form.